# The Role of Diagnostic Biomarkers, Omics Strategies, and Single-Cell Sequencing for Nonalcoholic Fatty Liver Disease in Severely Obese Patients

**DOI:** 10.3390/jcm10050930

**Published:** 2021-03-01

**Authors:** Charlotte W. Wernberg, Kim Ravnskjaer, Mette M. Lauridsen, Maja Thiele

**Affiliations:** 1Department of Gastroenterology and Hepatology, Hospital Southwest of Jutland, 6700 Esbjerg, Denmark; Charlotte.Wilhelmina.Wernberg@rsyd.dk (C.W.W.); Mette.Enok.Munk.Lauridsen@rsyd.dk (M.M.L.); 2Center for Functional Genomics and Tissue Plasticity (ATLAS), University of Southern Denmark, 5230 Odense, Denmark; ravnskjaer@bmb.sdu.dk; 3Department of Biochemistry and Molecular Biology, University of Southern Denmark, 5230 Odense, Denmark; 4Center for Liver Research, Department of Hepatology and Gastroenterology, Odense University Hospital, 5000 Odense, Denmark; 5Institute for Clinical Research, University of Southern Denmark, 5230 Odense, Denmark

**Keywords:** bariatric surgery, NAFLD, NASH, fatty liver, biomarkers, omics technologies

## Abstract

Liver disease due to metabolic dysfunction constitute a worldwide growing health issue. Severe obesity is a particularly strong risk factor for non-alcoholic fatty liver disease, which affects up to 93% of these patients. Current diagnostic markers focus on the detection of advanced fibrosis as the major predictor of liver-related morbidity and mortality. The most accurate diagnostic tools use elastography to measure liver stiffness, with diagnostic accuracies similar in normal-weight and severely obese patients. The effectiveness of elastography tools are however hampered by limitations to equipment and measurement quality in patients with very large abdominal circumference and subcutaneous fat. Blood-based biomarkers are therefore attractive, but those available to date have only moderate diagnostic accuracy. Ongoing technological advances in omics technologies such as genomics, transcriptomics, and proteomics hold great promise for discovery of biomarkers and increased pathophysiological understanding of non-alcoholic liver disease and steatohepatitis. Very recent developments have allowed for single-cell sequencing and cell-type resolution of gene expression and function. In the near future, we will therefore likely see a multitude of breakthrough biomarkers, developed from a deepened understanding of the biological function of individual cell types in the healthy and injured liver.

## 1. Non-Alcoholic Fatty Liver Disease in Severely Obese Patients

Obesity is a worldwide growing epidemic. A 2019 Lancet Commission declared obesity as one out of three epidemics that needs to be addressed due to their preeminent threat to human health [1]. Nearly one-quarter of U.S. citizens are predicted to be severely obese (body mass index, BMI, >35 kg/m^2^) by 2030, and it is estimated that the most frequent BMI-group among women in general will be ‘severely obese’ [2]. Non-alcoholic fatty liver disease (NAFLD) is a common manifestation of obesity, and the most common chronic liver disease worldwide, with a prevalence of 25% among adults [3]. NAFLD is a particular problem for severely obese patients, evidenced by observations in patients eligible for bariatric surgery. Up to 93% of these patients have NAFLD, one-fourth to two-thirds have non-alcoholic steatohepatitis (NASH), the progressive form of NAFLD, and 9–25% have fibrosis, scarring of the liver, which ultimately may lead to cirrhosis (Table 1) [4,5].

We do not have a complete overview of the molecular mechanisms that drive NAFLD towards NASH and liver fibrosis, but believe that NASH marks a high risk of fibrosis formation, when the inflammatory activity leads to hepatic stellate cell activation, with subsequent deposition of fibrillar collagens in the extracellular matrix [7]. However, fibrosis progression in NAFLD is asymptomatic and takes place over several decades, with an estimated one-third of NAFLD patients exhibiting fibrosis progression, and progression rates of 14 years per one fibrosis stage for patients with simple steatosis, versus 7 years for patients with NASH [8]. This demonstrates that NAFLD is a complex trait, where the inter-individual variation in disease phenotype likely arises from an interplay between genetic predisposition, environmental factors, and components of the metabolic syndrome [9,10]. For this reason, we still lack valid non-invasive alternatives to a liver biopsy. Fortunately, emerging technologies have made it possible to explore the complex biological systems in the liver during chronic liver injury [11,12]. Consequently, we are rapidly deepening our understanding of the pathways that drive NASH and liver fibrosis, while revealing new biomarkers and drug targets.

This review focuses on NAFLD in severely obese patients. We consider three main topics: Current state-of-the-art diagnostic tools, upcoming biomarkers in the form of new omics technologies, and finally, a probable new era in our understanding of liver disease aided by cutting edge technologies such as single cell sequencing.

## 2. Existing Diagnostic Tools for NAFLD in Severely Obese Patients

The diagnosis of liver fibrosis is central to risk stratification and management of NAFLD in severely obese patients, with advanced fibrosis (≥F3) being the major prognostic predictor [13]. Underdiagnosis of advanced fibrosis may lead to delayed treatment, with excess liver-related morbidity and mortality; while overdiagnosis causes futile investigations and patient concern [14,15]. The association between NASH and fibrosis progression makes it highly desirable to also identify NASH non-invasively. However, there are no currently well validated tools for NASH diagnosis, which today can only be diagnosed by a liver biopsy. Steatosis is not of prognostic value, but diagnosis and monitoring of steatosis may be valuable for evaluating the efficacy of interventions [16].

Liver biopsy remains the gold standard to stage fibrosis, grade hepatic inflammatory activity, and score steatosis. Liver biopsy is however prone to complications, sampling error, and observer-variance. In severely obese patients, liver biopsy is particularly problematic, because it may be difficult to obtain sufficient material with a standard percutaneous approach due to a large subcutaneous fat layer. Therefore, non-invasive diagnostic tests are highly needed, to replace liver biopsy as the regulatory surrogate endpoint in treatment trials [17].

Current non-invasive diagnostic tests can be divided into imaging, elastography, and blood-based markers (Table 2).

### 2.1. MR-Based Techniques

Magnetic resonance elastography (MRE) is the most accurate non-invasive fibrosis marker in NAFLD, with areas under the receiver operating characteristics curve (AUROCs) above 0.90 for both significant (≥F2) and advanced fibrosis (≥F3) [18]. During MRE, an acoustic driver placed on the patient’s abdominal right upper quadrant induces shear waves in the liver. The velocity of these waves can be converted to an elastogram, measuring liver stiffness [19]. MRE is not influenced by a large subcutaneous layer, and the method retains an AUROC exceeding 0.90 in severely obese patients [20]. However, most MR equipment has weight restrictions (140–180 kg/300–400 Ibs) and the diameter of the scanner, need for physical breath holding, and psychological challenges of cramp spaces can be a hindrance for morbidly obese patients. Another obvious limitation of MRE is restricted availability, reserved for tertiary centers and research settings. As all elastography techniques, MRE is also limited by false positives in case of hepatic inflammation, congestion, or obstructive cholestasis. Finally, diagnostic cut-offs may vary due to spectrum bias in various studies, and differences in the hardware and software used.

Magnetic resonance imaging proton density fat fraction (PDFF) is highly accurate for non-invasive assessment of hepatic steatosis, also in obese patients (AUROCs > 0.95) [21,22]. However, most diagnostic studies report mean BMI’s around 35 kg/m^2^ and only investigate per protocol diagnostic accuracies, discounting failed measurements. In contrast, a study specifically in patients undergoing bariatric surgery with a mean BMI of 45 kg/m^2^ reported failed measurements in more than one-third of patients with resulting poor accuracies in intention-to-diagnose analyses [23].

### 2.2. Ultrasound and Computer Tomography

Neither ultrasonography or computed tomography can be used to stage fibrosis [24]. Both modalities can detect cirrhosis in case of definite, radiological signs such as lobulized liver surface, irregular parenchymal structure and signs of portal hypertension, but ultrasonography will often be the preferred first-line imaging modality, because it is radiation-free and have Doppler.

Ultrasound also remains the most common way of diagnosing moderate and severe hepatic steatosis; it is accessible, safe, low cost, and has a good accuracy for detecting steatosis if ≥ 20% of hepatocytes contain fat vacuoles [25]. However, ultrasound has limited quality in patients with high BMI, leading to poor sensitivities and specificities in NAFLD cohorts of bariatric surgery patients, between 49–65% and 75–90%, respectively [26,27].

### 2.3. Ultrasound Elastography and Controlled Attenuation Parameter

Ultrasound elastography can be either vibration-controlled transient elastography (TE) with the equipment FibroScan (Echosens, Paris, France), point shear-wave elastography (pSWE), or 2-dimensional shear-wave elastography (2D-SWE) [28]. All techniques estimate liver stiffness by measuring the velocity of shear-waves induced in the liver by either a mechanical push-pulse (TE) or by the ultrasound beam (pSWE, 2D-SWE). The diagnostic accuracies for fibrosis in NAFLD patients are almost similar across the three techniques, with pSWE having slightly lower AUROCs than TE or 2D-SWE (AUROCs just above 0.80 for significant fibrosis, and above 0.85 for advanced fibrosis) [29]. However, reported diagnostic accuracies may be falsely high in the population of NAFLD patients because the quality and success rate of ultrasound elastography is hampered by a thick subcutaneous layer, and most studies report per-protocol, not intention-to-diagnose analyses [29]. The Fibroscan equipment has partially solved the problem of failed measurements by introducing the XL-probe for TE in patients with BMI > 30 kg/m^2^ or a skin-capsule distance exceeding 25 mm. Liver stiffness measurements with XL probe is approximately 20% lower than with the M probe in direct comparisons, but when using the M and XL probes as recommended, liver stiffness measures are comparable across fibrosis stages [30,31]. Consequently, the same diagnostic cut-offs can be used with both probes. A highly accurate cut-off to rule out advanced fibrosis in NAFLD is 8 kPa, while 12–15 kPa rules in advanced fibrosis with good accuracy, if causes of false positive measurements can be excluded [19,32].

Controlled attenuation parameter (CAP) is a tool for diagnosing steatosis only available with FibroScan. However, a recent individual-patient data meta-analysis found poor diagnostic accuracies for CAP to detect steatosis, with AUROCs well below 0.80 [33]. In the subpopulation of bariatric surgery patients, diagnostic accuracies were marginally better, but remained <0.80, with sensitivities and specificities below 80% and wide confidence intervals for the optimal cut-off points.

### 2.4. Blood-Based Markers

Diagnostic serum markers have an applicability advantage over imaging-based methods, in that it is almost always feasible to sample blood. The enhanced liver fibrosis test (ELF, Siemens Healthcare, Erlangen, Germany) is the most extensively investigated of the commercial serum markers [34]. ELF appears to have a slightly higher AUROC, than other commercial markers, but a direct comparison with FibroMeter^V2G^ did not show a difference in diagnostic accuracy [35]. ELF has slightly lower diagnostic accuracies (AUROC of 0.83 for advanced fibrosis) than TE when comparing studies using per-protocol analyses, while a diagnostic study in alcohol-related liver disease found comparable diagnostic accuracies of ELF versus elastography techniques in intention-to-diagnose analyses [36]. ELF seem to be equally accurate in patients with severe obesity, though existing studies are scarce and underpowered [37,38]. Other available commercial biomarkers like FibroMeter^V2G^, FibroTest/Fibrosure, Hepascore, and ProC3 may be good alternatives to ELF [39,40].

Non-patented biomarkers are algorithms from routine liver blood tests and clinical parameters [39]. They may be useful as first line testing for screening strategies, but has insufficient accuracy as diagnostic tools for significant and advanced fibrosis [41]. Similarly, a number of algorithms for steatosis assessment have been developed, utilizing biochemistry and clinical variables, but none of them perform with adequate diagnostic accuracy [19].

Since none of the available blood-based biomarkers are highly accurate, there is still an unmet need for biomarkers to diagnose advanced fibrosis. Current efforts focus on combining biomarkers, either in parallel or sequential [19]. Further, we need biomarkers to detect NASH, prognosticate, and to assess efficacy of interventions. For the latter, several candidate biomarkers are under evaluation [42,43]. Patients listed for bariatric surgery also constitute a particularly interesting cohort for evaluation of novel biomarkers, due to the easy access to peroperative liver tissue, and their massive, sustained weight loss post-surgery [44].

## 3. Omics Technologies as Upcoming Biomarkers

‘Omics’ refer to global disciplines in biological research such as genomics, transcriptomics, proteomics, or metabolomics. The goal of omics is to extract patterns and biological meaning from large-scale, high-dimensional data [45]. Today, several omics disciplines are well-established. In this section, we will discuss three omics disciplines that have already contributed significantly to our understanding of NAFLD (Figure 1).

### 3.1. Genomics

The disposition for NAFLD has been excessively investigated in genome-wide association studies (GWAS), whereby hundreds of thousands of single nucleotide polymorphisms (SNPs) are paired with information on liver diagnoses, liver enzymes, or other measurable traits [10,46,47]. Since GWAS study associations, they may inform NAFLD patient risk assessment and stratification, whereas use of genetic information for NAFLD treatment or pathophysiological understanding will require further functional analyses. Moving beyond simple associations, current challenges are the multifactorial nature of NAFLD pathogenesis, limited linkage between the traits investigated, and the causative SNPs, as well as the incomplete characterization of the 98.8% non-coding part of the human genome [48].

The strongest genetic risk factor for NAFLD is found in the PNPLA3 (patatin-like phospholipase domain-containing protein 3) gene. The PNPLA3 rs738409[G] allele (causing an I148M substitution) associates with both steatosis, NASH, cirrhosis, and hepatocellular carcinoma, and predicts an earlier age at NAFLD diagnosis [46,49,50,51]. Homozygous carriers of PNPLA3 rs738409[G] have a three-fold risk of steatohepatitis, and a four-fold risk of cirrhosis relative to non-carriers. Its association is most pronounced in Hispanics, indicating interactions with other genomic loci, and more than 20% of the population carries the risk allele, so the collective effect of PNPLA3 on NAFLD is high [51]. 

To date, further >10 genetic variants have been associated with NAFLD [46,52]. The best validated are TM6SF2 (transmembrane 6 superfamily member 2), which may be required for normal VLDL secretion, while GCKR (glucokinase-regulator) regulates hepatocyte glucose metabolism. Impaired TM6SF2 and GCKR function associates with fibrosis and NAFLD, but not NASH [46]. HSD17B13 [TA] (hydroxysteroid 17-beta dehydrogenase 13) encodes a hepatic lipid droplet protein and is associated with a protective effect against cirrhosis in fatty liver diseases due to NAFLD or alcohol [53]. 

Adiposity amplifies the effects of the PNPLA3 allele along the entire spectrum of NAFLD severity [10], but it is unknown whether severe adiposity amplifies or attenuates the effect of the other SNPs. In subjects eligible for bariatric surgery, PNPLA3 and GCKR were associated with steatosis and fibrosis [54,55]. However, both studies included patients without advanced fibrosis, and used surgical biopsies, which are obtained preoperatively after weight loss, when steatosis has regressed.

From a biomarker perspective, attempts have been made to incorporate the validated genetic risk alleles in genetic risk scores, for use as risk stratification tools [56]. So far, none have passed to clinical implementation, and their cost–benefit, utility, and context of use as required by regulators are still up for debate. However, with diminishing costs for targeted SNP sequencing, we expect that genetics may be an integrated part of patient management in few years.

### 3.2. Transcriptomics

Transcriptomics is the quantitative assessment of all RNA, coding as well as non-coding, and it offer insights into differential gene expression and gene regulatory mechanisms in a cell population or tissue. Bulk liver transcriptomics were the first to investigate NAFLD, and are still the most common, but are disproportionately dominated by the parenchymal hepatocytes, which make up 60–70% of the cells in the healthy liver (Figure 2). Bulk transcriptomics therefore has limited value for the analysis of less abundant cell types [57].

Bulk transcriptomics studies in human NAFLD and steatohepatitis have described changes to genes involved in lipid handling, inflammation, cell migration, extracellular matrix turnover, and regenerative processes [58,59]. Genes linked to core hepatocyte functions—the metabolism of lipids, glucose, amino acid, and xenobiotics—appear down-regulated [59,60,61], while up-regulated genes in severe NAFLD are associated with tissue remodeling, progenitor cells, cancer, and cardiovascular disease [58]. A recent study of 620 severely obese patients compared transcriptomes of biopsy-validated simple steatosis versus NASH, and revealed distinct gene sets indicative of sexual dimorphism, implicating that novel NASH drug targets could be gender dependent [62]. Immune activation can also be observed on a transcriptional level: A study in severely obese patients with NASH and/or significant fibrosis found interleukin 32 to be the most heavily upregulated transcript, and circulating interleukin 32 levels corresponded to the expression in the liver tissue [63]. This finding was replicated in a recent multicenter study of 206 NAFLD patients, together with an additional 24 gene signatures, most of whom could be translated to circulating protein levels [61]. These findings underline the particular strength of transcriptomics, linking changes in gene expression to serum markers.

### 3.3. Proteomics

The proteome reflects immediate cellular processes, and therefore holds obvious potential for biomarker and drug-target discovery. Mass spectroscopy-based proteomics now makes it possible to identify and quantify thousands of proteins from liver tissue and the circulation [64]. This is in contrast to prior biomarker-finding studies, which targeted a limited number of specific proteins [34].

The first proteomics study in NAFLD determined hepatic protein abundance in liver samples from obese subjects in four groups: Obese without NAFLD, simple steatosis, and NASH with or without fibrosis [65]. They identified nine proteins that were differentially expressed across the four groups, many with probable pathophysiological functions: Hepatic lipid content, inflammation, and fibrosis. Lumican was one of the overexpressed proteins in NASH, an extracellular matrix proteoglycan that regulates collagen fibril-formation. Another case-control study even found elevated serum levels of lumican in NASH [66]. They found additional differences in 55 circulating proteins between simple steatosis and NASH, and differences in 15 proteins between NASH without versus with advanced fibrosis. Proteins involved in platelet aggregation and coagulation were elevated in NAFLD and NASH patients, while fibrinogens were significantly reduced. Finally, a case control study of the plasma proteome described nine candidate biomarker proteins, of which polymeric immunoglobulin receptor increased in persons with NAFLD, and further with cirrhosis. [67]. This highlights the potential for proteome analysis to discover new biomarkers in liver disease. 

Another highly promising omics field is metabolomics, including circulating, urine or stool lipids, and metabolites [68,69,70]. This field is particularly interesting due to the causal role of metabolic dysfunction for the progression liver fibrosis in NAFLD patients. As a consequence, diagnostic molecules may also yield pathophysiological understanding and be druggable targets [71]. However, both the proteomics and metabolomics fields need to validate candidate biomarkers in biopsy-verified, diagnostic test cohorts that represent the full spectrum of disease in consecutively recruited NAFLD patients.

## 4. Single-Cell and Cell Type-Resolved Omics Approaches to NAFLD

The liver tissue has a complex lobular architecture and is composed of multiple cell types that interact with one another and communicate with surrounding tissues (Figure 2 and Figure 3).

By enabling the characterization of individual cell types during NAFLD progression, single-cell approaches have already offered novel insights into hepatocellular dynamics that could not be obtained from conventional bulk assays [61,72]. The techniques are still reserved for basic science, with few translational studies, but further technological developments and cost-reductions will likely spread the use of single-cell and cell type-resolved omics approaches, for deepened understanding of the cellular processes during NAFLD progression and regression, and eventual biomarker development.

### 4.1. Single-Cell Transcriptomics

Single-cell RNA sequencing and single-nucleus RNA sequencing allowed scientists to characterize transcriptomes of individual cells in human and rodent liver, to establish a cell-atlas of the human liver, and to understand the functions and crosstalk of hepatic cell types [73]. Though most evidence is from murine models, and study normal liver, liver cancer, or cirrhosis, transcriptomic studies of liver biopsies from NAFLD patients have already offered a number of promising biomarkers [74]. 

In a study of healthy liver tissue, MacParland et al. described 20 discrete cell populations of hepatocytes, endothelial cells, cholangiocytes, hepatic stellate cells, B cells, conventional and non-conventional T cells, NK-like cells, and distinct intrahepatic monocyte/macrophage populations [75]. They showed with single-cell resolution that the gene expression of human hepatocytes, as well as non-paranchymal cells, reflect their position along the porto-central axis in keeping with functional zonation of the lobule. Another single-cell study of human liver further refined the concept of hepatocyte and endothelial cell co-zonation, to suggest cooperation across cell types around essential liver functions [76].

In the first single-cell-resolved study of human liver cirrhosis, Ramachandran et al. compared five healthy liver tissue patients with five cirrhosis patients. The authors identified TREM2+/CD9+ scar-associated macrophages, ACKR1+/PLVAP+ endothelial cells, and PDGFRα+ collagen-producing myofibroblasts specifically associated with cirrhosis and spatially restricted to the fibrotic niche [12]. A later study confirmed that PLVAP (plasmalemma vesicle-associated protein) is expressed by hepatic stellate cells, and its expression suppressed upon liver injury in NASH [77]. The same study identified core hepatic stellate cell genes, whose expression proved highly predictive of advanced fibrosis in NASH patients. This highlights the importance of biomarkers that focus on non-parenchymal liver cells, as they are the main actors in the fibrogenic response to liver injury. Similarly, a number of recent single-cell studies of murine NASH models have shed light on hepatic macrophage dynamics and the inflammatory response in NASH livers where macrophages gradually populate the hepatic niche replacing Kupffer cells upon NASH progression, and some assume the scar-associated TREM2+ phenotype described above [78].

### 4.2. Cell-Type Resolved Proteomics

Improvements in mass spectroscopy have made it possible to obtain comprehensive proteome descriptions of individual cell types, stopping short of actual single-cell proteomics [79]. Azimifar et al. purified hepatic cell types from male mice and identified 11,520 proteins to construct a cell-type-resolved atlas of the mammalian liver proteome [80]. They showed that the 100 most abundant proteins comprise over 40% of the total mass of the proteomes. Many of these proteins were metabolic enzymes, stressing that the liver is a metabolic hub of the body. However, one of the first cell-resolved studies of the human liver proteome revealed large discrepancies between human and murine proteomes [81]. They used three healthy liver transplant donors to perform quantitative proteomics analysis on the major cell lines: Hepatocytes, endothelial cells, Kupfer cells, and hepatic stellate cells. From >9700 identified proteins, 53% (>5100) were found in all cell types, indicating mutual basal cellular processes. 

While actual single-cell proteomics of human livers have not been done yet, recent technological advances has opened up for large-scale proteomics in formalin-fixed, paraffin-embedded liver biopsy tissue, with great promise for further understanding of hepatic cells’ functional contributions to liver biology [82].

## 5. Conclusions

Current diagnostic tools in NAFLD are of limited use for patients with severe obesity due to lack of technical applicability, moderate diagnostic accuracies for advanced fibrosis, and absence of validated biomarkers for other important indications such as diagnosis of NASH, prognostication, monitoring, and efficacy-of-intervention. Fortunately, technological advances progressively allow for large-scale, high-throughput omics analyses. As a consequence, genomics, transcriptomics, and proteomics will, in the coming years, move from basic and translational science into biomarker development and validation.

## Figures and Tables

**Figure 1 jcm-10-00930-f001:**
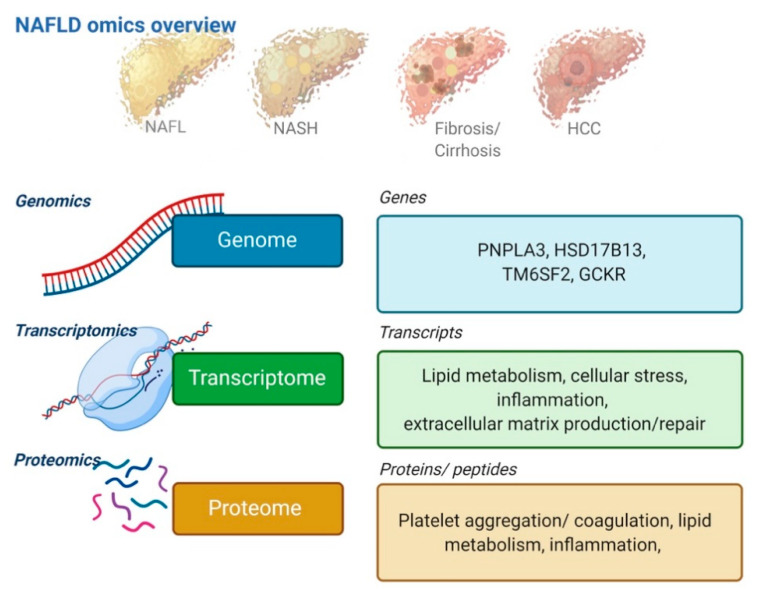
Omics technologies which may mature into implementable biomarkers in the near future; with examples of signals from genomics, transcriptomics, and proteomics that have been associated with NAFLD disease severity.

**Figure 2 jcm-10-00930-f002:**
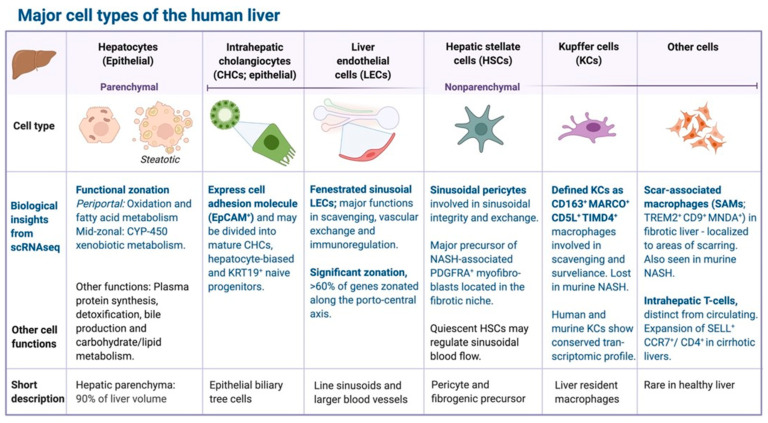
Major cell types of the human liver. Single-cell and cell type-resolved omics studies have all added to our understanding of liver biology in health and disease.

**Figure 3 jcm-10-00930-f003:**
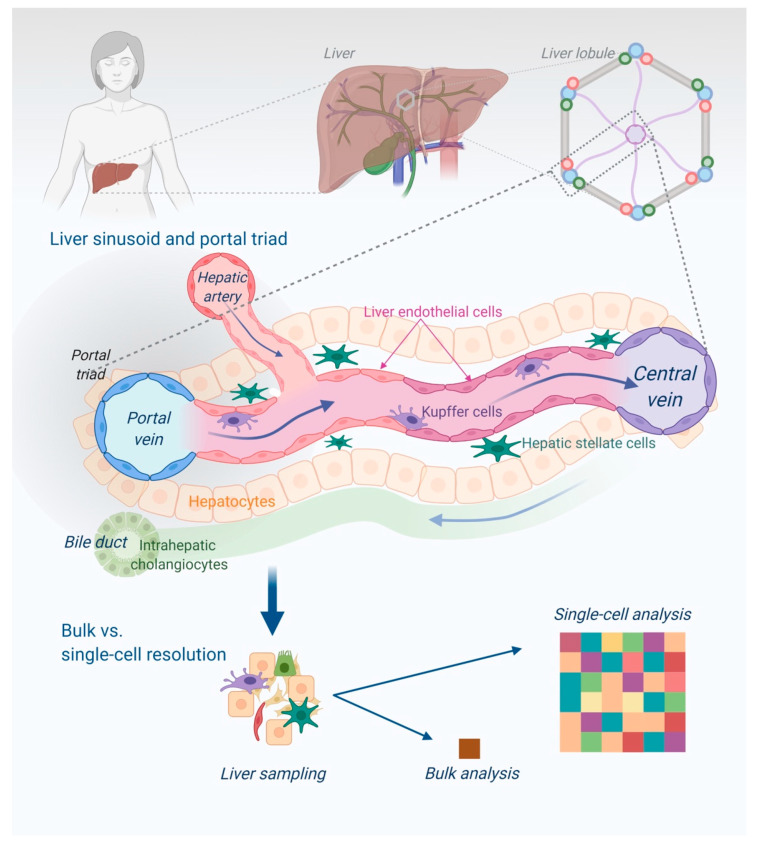
The structure of a liver lobule, its zonation and different cell types highlights the advantage of single-cell analysis. Healthy hepatic lobules are zonated from the portal triads (portal vein, hepatic artery, and bile duct) towards the central vein. Oxygenated, arterial blood and nutrient-rich portal venous blood enters the lobule from the portal triad, and is transported back to the circulation via a central vein.

**Table 1 jcm-10-00930-t001:** Histological staging and grading of non-alcoholic fatty liver disease ^1^.

Histological Characterisation	Description
Steatosis	When more than five percent of hepatocytes contain fat vacuoles. Scored according to degree of fat infiltration: S1: Minimal, 5–33% hepatocytes infiltrated by fat. S2: Moderate, >33–66% hepatocytes infiltrated by fat. S3: Severe, >66% hepatocytes infiltrated by fat.
Non-alcoholic steatohepatitis	Defined by presence of both steatosis, ballooning, and lobular inflammation. Activity is scored according to severity: Few ballooned hepatocytes versus prominent ballooning; and <2 inflammatory foci per 200Xfield, 2–4 foci, or >4 foci.
Fibrosis	In NAFLD, fibrosis begins by pericellular deposition of fibrillar collagen fibers; gradually expanding to form large fibrotic septae. Fibrosis is staged according to distribution and magnitude: F1: Mild, perisinusoidal or periportal fibrosis. F2: Moderate, perisinusoidal and portal/periportal. F3: Severe, bridging fibrosis. F4: Cirrhosis, characterised by regeneration nodules.

^1^: Described according to the most commonly used histology score for grading and staging of NAFLD, the NAS-CRN system and the Kleiner fibrosis score (Nonalcoholic Steatohepatitis Clinical Research Network) [6].

**Table 2 jcm-10-00930-t002:** Advantages and disadvantages of diagnostic markers for the assessment of steatosis and advanced fibrosis in severely obese patient with non-alcoholic fatty liver disease (NAFLD).

	Fibrosis		Steatosis	
	Advantages	Disadvantages	Advantages	Disadvantages
Imaging				
Ultrasound	Low cost, widely available in primary and secondary care	Poor quality in severely obese patients. Only accurate in case of late-stage cirrhosis	Low cost, widely available in primary and secondary care	Poor quality in severely obese patients. Only accurate if >20% fat-infiltrated hepatocytes
CT	Widely available in hospital care	Radiation. Only accurate in case of cirrhosis	Widely available in hospital care	Radiation. Only accurate if >20% fat-infiltrated hepatocytes
MRI	No radiation in contrast to CT	Low availability. Only accurate in case of cirrhosis	MRI-PDFF is the most accurate non-invasive marker of steatosis with AUROC’s > 0.90	Low availability. Severely obese patients may need special scanner
Elastography				
TE	Available in most hepatology clinics. The XL probe is developed for obese patients	Moderate accuracy with AUROC’s 0.80–0.85	Controlled attenuation paramenter, a non-invasive steatosis measure, is available together with TE	Poor diagnostic accuracies with AUROC’s < 0.80
pSWE	Available as complementary software on many ultrasound equipment	High risk of unreliable measures in severely obese patients	-	-
2D-SWE	Measures liver stiffness in a larger region of interest than pSWE and TE	High failure rate in severely obese patients	-	-
MRE	Most accurate non-invasive marker of fibrosis, with AUROC’s > 0.90	Low availability. Severely obese patients may need special scanner	-	-
Blood based				
ELF	Can be sampled in primary care	Patented test. Moderate accuracy with AUROC’s 0.80–0.85	-	-
FIB-4, APRI and NFS	Can be measured from routine liver blood tests	Insufficient diagnostic accuracy with AUROC’s < 0.80	-	-

Abbreviations and explanation. 2D-SWE: Two-dimensional shear-wave elastography, an ultrasound-based technique. AUROC: Area under the receiver operating characteristics curve. CT: Computer tomography. ELF: Enhanced liver fibrosis test, an algorithm of hyaluronic acid, propeptide of type III collagen, and tissue inhibitor of metalloproteinase-1. FIB-4: Fibrosis-4 test, an algorithm of age, aspartate transaminase, alanine transaminase, and platelet count. NFS: NAFLD Fibrosis Score, an algorithm of age, body mass index, presence of diabetes, aspartate transaminase, alanine transaminase, platelet count, and albumin. MRE: Magnetic resonance elastography. MRI: Magnetic resonance imaging; PDFF: Proton density fat fraction. pSWE: Point shear-wave elastography, an ultrasound-based technique. TE: Vibration controlled transient elastography, an ultrasound-based technique.

## Data Availability

No new data were created or analyzed in this study. Data sharing is not applicable to this article.

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
