# Peer review of "The Role of Diagnostic Biomarkers, Omics Strategies, and Single-Cell Sequencing for Nonalcoholic Fatty Liver Disease in Severely Obese Patients"

_jcm, 2021, doi:10.3390/jcm10050930_

Round 1
Reviewer 1 Report
Wemberg et al. proposed a review integrating different methods to assess NASH and fibrosis in patients with NAFLD.
The manuscript is interesting, well-written.
I only have got some limitations:
- It is difficult to tell that ELF is the most accurate score and, some lines after, that there was no comparison. Furthermore, comparisons were proposed by Angers' team.
- Combinations between non-invasive tests were performed, it would be interesting to cite them.
- I would be interesting in CK18 role. Is it voluntar ?
Reviewer 2 Report
I read with interest the review of Wenberg et al regarding the biomarkers for NAFLD in patients with severe obesity. This review reflects the drawbacks of currently available diagnostic biomarkers for fibrosis in patients with NAFLD and severe obesity, and the promising future of omics strategies and single-cell sequencing for discovery of biomarkers. There are a few issues that need to be addressed.
- Suggest you delete Table 1 since it is known information to any clinician and researcher interested in NAFLD.
- Discuss in section 1 why this review is focusing on obesity and not diabetes, for example, since it is another factor frequently associated with NAFLD.
- P3, line 81: Add the limitations of biopsy not only for obese patients (complications, sampling error, inter- and intra-observer variability).
- Table 2: I advise that you include MRI-PDFF, FLI and HSI for hepatic steatosis, and APRI and Hepamet Fibrosis Score for liver fibrosis.
- P5, line 143: Discuss the meta-analysis by Karlas T et al (J Hepatol 2017) where only studies with M probes are analyzed showing an AUROC greater than 0.8. In the meta-analysis by Petroff, probe XL was used in approximately 45% of the patients. This reinforces the limitation of these non-invasive methods in obese patients.
- P5, line 166: Missing references.
- P5, line 163: Suggest you mention that these scores have a high negative predictive value, especially FIB-4, which is important for a screening method. However, the principal drawback of all these biomarkers is that none of them is specific of the liver and their results can be influenced by co-morbidities of patients, so a critical interpretation of the result is necessary. Also discuss the Hepamet Fibrosis Score which seems not to be affected by BMI (Ampuero J, Clin Gastroenterol Hepatol 2020).
- I advise that you mention something about metabolomics.
Author Response
Please see attachement

Reviewer 3 Report
Dear Authors
This reviewer would like to congratulate you on this effort to address an important topic, but he/she has major criticism.
In Table 2.....it is very surprising that a great percentage of severely obese patients could have less than 20% of steatosis...reason for which the disadvantage for diagnosing NAFLD at US in this specific population is of relative importance, at the light that in these patients steatosis was found in 86% of total cases, as evident in.... Liver pathology and the metabolic syndrome X in severe obesity. J Clin Endocrinol Metab. 1999 May;84(5):1513-7.
Authors should expand the biomarkers utility, very useful mainly in epidemiological studies where other approaches are far more expensive and scarcely available. In fact the main distinction between the reliability of surrogates lies on their field of application (research/trials/epidemiology).
For example, can we genetically test every obese patient for NAFLD presence, at the light that obesity hits about one third of the population?...as evident in .....N C Med J. 2013 Nov-Dec; 74(6): 530–533.
In fact, in your study a cost/benefit analysis is mandatory for every proposed diagnostic approach as well as a right comparison between them.
It is correct to state that we we do not have, at least until now, clear the mechanisms inducing and/or worsening NAFLD, and for this valid alternatives to liver biopsy are lacking, as well as efficient therapies as evidenced in...J. Clin. Med. 2020, 9(1), 15.
Among others surrogates of NAFLD that should be clearly mentioned and discussed such as FLI and HSI, the new triglycerides/glucose ratio is a great novelty as per....Triglyceride Glucose Index Is Superior to the Homeostasis Model Assessment of Insulin Resistance for Predicting Nonalcoholic Fatty Liver Disease in Korean Adults. Endocrinol Metab (Seoul). 2019 Jun;34(2):179-186.
Author Response
Please see attachement

Round 2
Reviewer 3 Report
Authors answered comments
Author Response
Thank you,